# Allograft and Autologous Reconstruction Techniques for Neglected Achilles Tendon Rupture: A Mid-Long-Term Follow-Up Analysis

**DOI:** 10.3390/jcm12031135

**Published:** 2023-02-01

**Authors:** Cristina Jiménez-Carrasco, Fadi Ammari-Sánchez-Villanueva, Estefanía Prada-Chamorro, Antonio Jesús García-Guirao, Sergio Tejero

**Affiliations:** 1Orthopedic Surgery and Traumatology Service, Hospital San Juan de Dios del Aljarafe, 41930 Bormujos, Spain; 2Orthopedic Surgery and Traumatology Service, Hospital Universitario Virgen del Rocío, 41013 Seville, Spain; 3School of Medicine, Universidad de Sevilla, 41009 Sevilla, Spain

**Keywords:** Achilles allograft, Achilles chronic rupture, Achilles reconstruction, Achilles tendon, neglected Achilles rupture

## Abstract

Achilles tendon ruptures that are not immediately recognized and treated are sometimes diagnosed as delayed injuries and may require different surgical repair options based on gap size. The potential complications associated with using an allograft for reconstruction may lead some surgeons to prefer the use of autologous techniques. However, allografts are often considered a salvagement option when large defects are present. In this study, we examined the long-term clinical outcomes and complications of 17 patients who underwent surgical repair for chronic ruptures with large gaps using both autologous and allograft techniques. During an 11-year period, nine patients were treated with autologous techniques (mean gap of 4.33 ± 1.32 cm) and Achilles allograft reconstruction was performed in eight patients (47.1%) (mean gap of 7.75 ± 0.89 cm). At a mean of 82 ± 36.61 months of follow-up, all 17 patients (100%) were able to perform a single heel rise and improved AOFAS (American Orthopaedic Foot and Ankle Society) and ATRS (Achilles Tendon Total Rupture Score) scores. No infections, complications, or re-ruptures were recorded at the end of the follow-up. No significant differences were found in the AOFAS and ATRS scales between both techniques. When an extensive defect is present, the reconstruction with an Achilles tendon allograft can be considered a proper treatment option, as it does not show a higher rate of complications than autologous techniques achieving similar functional outcomes.

## 1. Introduction

The Achilles tendon, the largest and strongest tendon in the human body, plays a crucial role in the movement of the foot and ankle [1]. Despite its strength, it is one of the most commonly ruptured tendons, particularly in middle-aged individuals who occasionally participate in sports activities [2]. Achilles tendon tears can occur due to various risk factors, including medical treatment, genetic and systemic conditions, and biomechanical stress. However, the exact cause remains controversial as there is limited strong evidence supporting any one factor [3]. The treatment of acute Achilles tendon ruptures has been studied extensively, with both surgical and non-operative options available. A combination of minimally invasive surgery with early rehabilitation protocol is considered the overall best treatment strategy [4,5]. However, in the acute setting, surgical treatment has been found to not improve functional outcomes in the long-term follow-up, while non-operative treatment is associated with an increased risk of re-rupture [6]. Unfortunately, up to 25% of these acute injuries are misdiagnosed and present as delayed injuries [7], known as neglected chronic Achilles tendon ruptures. These chronic ruptures can cause chronic pain, claudication, and weak or absent heel rise, greatly affecting daily life. Most surgeons agree that chronic ruptures should be managed surgically, with the goal of restoring and maintaining the length and tension of the Achilles tendon, to enable propulsive gait through the gastrocsoleus muscle–tendon complex [8,9]. Due to the retraction of the tendon ends and replacement of the ruptured area with fibrous scar tissue, it can be challenging to repair the gap in an end-to-end manner, requiring the use of alternative techniques for surgical repair. Various methods have been proposed to achieve this goal, such as V-Y tendon plasty, tendon transfers, allograft and autograft reconstruction, and synthetic and biologic matrix augmentation. However, there is no standard treatment for neglected chronic Achilles tendon ruptures [2,10], and long-term follow-up results are insufficient. Therefore, choosing the optimal therapy among the various methods is often a challenge for orthopedic surgeons, and more research is needed to find the best approach for these complex cases. The purpose of the present study is to describe and compare the clinical outcomes and complications at long-term follow-up in chronic ruptures with large gaps, using either autologous techniques or Achilles allograft reconstruction.

## 2. Materials and Methods

### 2.1. Study Design

With the approval of the responsible ethical committee, a retrospective comparative observational study and outcome analysis was conducted at our institution, in order to analyze patients treated surgically for a chronic neglected Achilles tendon rupture. Demographic and perioperative data were collected including age, gender, metabolic disorders (obesity, hypercholesterolemia, diabetes mellitus, or thyroid disorders), intraoperative gap, type of surgery, and complications. As the primary goal of this study was to compare the different outcomes between both autologous and allograft techniques, patients were classified into two groups: (i) treated with V-Y advancement ± FHL (flexor hallucis longus) transfer, meaning “autologous” techniques, and (ii) treated with reconstruction with an Achilles tendon allograft.

### 2.2. Eligibility

An 11-year study period was set, from 1 January 2008 to 31 December 2019, ensuring a minimum follow-up of 24 months. The inclusion criteria comprised: (i) Achilles tendon rupture diagnosed more than 6 weeks after injury; (ii) inability to perform a single heel rise; (iii) a gap length greater than 2 cm diagnosed by magnetic resonance imaging (MRI); (iv) not being an open injury at first; (v) having at least 2 years of postoperative adequate follow-up data; (vi) age greater than 14 years. Cases due to failure of conservative treatment were excluded. No exclusions for underlying disease conditions were set. Within the observation period, 21 patients were identified, of which only 17 met the inclusion criteria. All patients had preoperative magnetic resonance imaging (MRI) scans available, and the gap greater than 2 cm between the tendon ends was measured for the preoperative planning. All surgical procedures were performed by orthopedic surgeons who are engaged in full-time foot and ankle practice.

### 2.3. Surgical Algorithm and Technique

The surgery was done under spinal anesthesia with a thigh pneumatic tourniquet in a prone position. An extensive midline approach to the Achilles tendon was used to expose the rupture. After resecting the nonviable dystrophic tendon ends, the true tendon gap was measured with the knee flexed at a 30º degrees angle and the ankle plantar flexed to 20 degrees. The surgical technique performed was then chosen depending on this measurement of the tendon rupture gap, according to the following algorithm, also described in Figure 1.

For a gap less than or equal to 2 cm, a direct end-to-end repair was performed; for a gap between 2 cm and 6 cm, an Achilles tendon advancement procedure (V-Y tendon plasty) was performed, associating an FHL tendon transfer to the calcaneus when the gap was between 4 cm and 6 cm (Figure 2); for a gap greater than 6 cm, a bone–tendon full-length Achilles allograft reconstruction was performed. In the last case, we directly insert the allogenic calcaneal bone block in a press-fit manner within the native (host) calcaneus, previously prepared with a socket, without any osteosynthesis or anchor, and a proximal tendon-to-tendon with absorbable suture.

Paratenon was sutured whenever possible, and skin closure was done using the technique of Allgöwer–Donati. A lower-leg anterior splint in resting equinus position was placed on in all cases. After three weeks, wound review and suture removal were performed if possible and an articulated walker boot was placed on, preventing dorsiflexion but allowing active plantar flexion for autologous reconstruction and passive plantar flexion for allograft reconstruction. Partial weight bearing is started (with crutches and a walker) at three weeks for autologous techniques and at 2 months for allograft techniques.

### 2.4. Outcome Assessment and Data Analysis

In both the global sample and subgroups, the primary outcome measure in the present study was the American Orthopaedic Foot and Ankle Society Ankle-Hindfoot Score (AOFAS_AH_) and The Achilles Tendon Total Rupture Score (ATRS). The ability of performing a single-heel-rise at the end of follow-up was also collected.

Demographic continuous data were presented as mean ± standard deviation (range). The normality of each variable was evaluated using the Shapiro–Wilk test. Scores (ATRS and AOFAS_AH_) were treated as nonparametric data and presented as the median score ± interquartile range. Categorical variables are expressed in percentages. Relationships between variables were analyzed using the $\chi^{2}$-test for qualitative variables. The nonparametric Mann–Whitney U test was used to compare quantitative non-normally distributed numerical data among the groups. Results were considered statistically significant at a *p* value < 0.05. The statistical analysis was carried out using the statistical software MedCalc^®^ version 19.8 for Windows^®^.

## 3. Results

### 3.1. Sample Characteristics

We evaluated 17 patients that fulfilled the inclusion criteria (14 men (82.4%) and 3 women (17.6%)), with a mean age of 42.8 ± 14.0 years (range 24–69 years). The mean follow-up was 82.18 ± 36.61 months (range 26–150 months). The mean intraoperative gap size after debridement was 5.94 ± 2.08 cm (range 3–9 cm). Nine patients (52.9%) were treated with autologous techniques (6 with isolated V-Y advancement (35.3%) and three with V-Y advancement associated with FHL transfer (17.6%)), with a mean gap of 4.33 ± 1.32 cm (range 3–6.5 cm). Reconstruction using allogenic Achilles tendon allograft was performed in eight patients (47.1%), with a mean gap of 7.75 ± 0.89 cm (range 7–9 cm). One patient in the allograft group associated an FHL transfer to the reconstruction. This case was secondary to a deep infection of a repair of an acute Achilles tendon rupture that, after a deep debridement, gentamicin-PMMA beads, and antibiotics, a sural flap and reconstruction were done after 12 weeks (second stage). This case is exposed in Figure 3. Subgroup and patient data are shown in Table 1 and Table 2.

### 3.2. Follow-Up and Complications

At a mean of 82 ± 36.61 months of follow-up (range 26–150), all 17 patients (100%) treated were able to perform a single heel rise. None of them were limited in performing activities and they achieved daily life activity levels without restrictions.

No infections or complications were recorded at the end of the follow-up. During the follow-up period, there were no instances of recurrent Achilles tendon rupture or allograft rupture among the patients.

### 3.3. AOFAS and ATRS Scores

In the full sample, the median AOFAS_AH_ score increased significantly (*p* < 0.001) from a preoperative median of 55 (46–67) to a postoperative median of 96 (92.5–100). The median ATRS score increased significantly (*p* < 0.001) from a preoperative median of 35 (24.5–40.5) to a postoperative median of 90 (79.75–95.25), at the end of the follow-up. In the autologous techniques group, the median AOFAS_AH_ score increased from a preoperative median of 55 (46–69) to a postoperative median of 96 (94–100). The median ATRS score in the autologous group increased from a preoperative median of 35 (33.75–38.5) to a postoperative median of 90 (85.5–95.25), at the end of the follow-up. In the allograft reconstruction group, the median AOFAS_AH_ score increased from a preoperative median of 50.5 (46.5–62) to a postoperative median of 95 (87–100). The median ATRS score in the allograft group increased from a preoperative median of 27 (19–45) to a postoperative median of 88 (79.5–94) at the end of the follow-up. These findings are shown in Table 3.

No significant differences between groups (autologous techniques and allograft reconstruction) were found in the AOFAS or ATRS scores.

## 4. Discussion

For most foot and ankle surgeons, chronic Achilles tendon ruptures with large gaps are challenging to treat [9,11]. Many different techniques can be used to repair the rupture and, in general, they tend to yield similar functional outcomes, as demonstrated by the improvement in functional outcome scores in most studies published to date [8,12,13,14,15,16]. In our study, we also found similar functional results at mid and long-term follow-up in both groups and did not observe more complications in the allograft group. We believe that this technique could be a viable option for gaps less than 6 cm in some circumstances, particularly when the priority is to avoid soft tissue damage from a long incision for a V-Y technique. Comparison of different techniques is difficult. The choice of surgical technique depends on factors such as the size of the gap, the quality of the tendon, and the surgeon’s comfort and expertise [8,13].

The V-Y tendon plasty was first introduced by Abraham and Pankovich [17] as an effective treatment for chronic Achilles tendon ruptures. Khiami et al. [18] suggested that the V-Y tendon plasty is suitable for defects of 2 to 5 cm, while McClelland and Maffulli [7] reported satisfactory results for gaps less than or equal to 6 cm. In our institution, we use the V-Y tendon advancement technique for gaps between 2–6 cm, and our significant improvement in the AOFAS and ATRS scores supports these conclusions. However, other studies have reported satisfactory outcomes with the V-Y advancement technique for gaps greater than 6 cm [9,11].

Many previous studies also reported good outcomes after treatment of chronic Achilles tendon rupture with isolated FHL tendon transfer or FHL transfer with additional augmentation (i.e., gastrocnemius fascial advancement flap or Achilles tendon turndown flap) [19,20,21]. FHL transfer was first described for this pathology in 1993 by Wapner [22], who identified the main advantages of the FHL tendon transfer relative to other tendon transfers (i.e., PB (peroneus brevis) or FDL (flexor digitorum longus) tendons). Yeoman et al. [23], Mohammed et al. [24], and Ole Kristian Alhaug et al. [25] reported good outcomes after treatment of chronic Achilles rupture with FHL tendon transfer in 11, 10, and 21 patients, respectively. The study of Oksanen et al. [21] showed 52% FHL muscle hypertrophy after FHL tendon transfer for the chronic Achilles tendon rupture, indicating a strong adaptation capacity of this muscle. Lin and colleagues [26] consider FHL transfer only when the tendon stumps at the calcaneus do not have enough integrity of the Achilles tendon. The survey by Villarreal et al. [27] demonstrated a tendency to use the FHL for the management of large chronic ruptures over other methods.

In our opinion, when the gap exceeds 4 cm, the vascularization of the elongated segment may be compromised, and the FHL muscle located just below it could serve as both a mechanical and biological support. As such, we use the association of V-Y advancement with FHL transfer for gaps between 4 and 6 cm, and perform only V-Y advancement when the defect is between 2 and 4 cm. However, isolated FHL transfer may be sufficient for patients with high age, low functional demands, or high morbidity. This theory is supported by Jirun A. [2], whose meta-analysis shows that FHL with additional augmentation technique results in a better postoperative AOFAS score compared to the isolated FHL transfer technique (including both open or endoscopic techniques) but without statistically significant results. In our study, we performed FHL transfer in the autologous group in three patients, achieving excellent results with low morbidity. The AOFAS collected results (median 96 points, mean 94.29 points) were similar to those published in the literature (mean 95.25 points) [2,15].

The Peroneus brevis (PB) tendon graft can also be used as an alternative for reconstruction in neglected Achilles ruptures. Maffulli et al. [7] reported their outcomes in 32 patients at a mean follow-up period of 48 months, using a less invasive reconstruction method and achieving good results. However, there are concerns with using a PB tendon transfer, such as a loss of eversion strength and the pull of the transferred PB tendon not matching the Achilles medial moment arm [14,28]. In our opinion, the PB tendon transfer should not be the first choice.

Other techniques have been proposed for treating chronic Achilles tendon tears, such as a minimally invasive reconstruction using an autologous hamstring graft [29], which has shown promising biomechanical and long-term clinical results in a study. Another technique is the use of an autologous quadriceps tendon graft [30] which is injected with platelet-rich plasma (PRP) and fixed to the bone with a small screw.

Allograft is another valid surgical choice for chronic Achilles rupture. Two types of tendon allograft have been reported in the literature: Achilles tendon allograft [12,31,32,33,34] and peroneus brevis tendon allograft [35]. Allograft use has certain drawbacks, including the potential for disease transmission, immune response, increased cost, and the need for the allograft to integrate with the host tissue [10,36]. However, it also has some advantages: mechanical properties, the avoidance of harvest site morbidity, the provision of adequate tissue quantity and quality, and decreased surgical time [36]. Nellas et al. [31] reported a case using an Achilles tendon allograft first in 1996, and 17 years later Hanna et al. [37] presented an Achilles tendon allograft with calcaneal bone block as an effective treatment for chronic Achilles tendon rupture. Our cohort of patients who underwent allograft reconstruction included six individuals with a follow-up period of over 56 months. This technique was used to repair large segmental defects in the Achilles tendon that measured greater than 6 cm, and it also incorporated a bone block for implantation in the calcaneus. All of them achieved an improvement in the AOFAS scores from a median of 55 preoperatively to 96 postoperatively. Results from other studies support this technique as an alternative for the reconstruction of chronic Achilles rupture [12,15,31,32,33,37]. To date, there is only one case series that compared the results of allograft with other reconstruction methods [12]. In this study, Park and colleagues [12] found that the mean ATRS score of allograft and autograft postoperatively were comparable (96 and 92.3, respectively). From our experience with the presented patients, the results are similar (the median AOFAS score of allograft and autograft postoperatively was 95 and 96, respectively). We cannot conclude that allograft reconstruction is a better choice because the differences between techniques were not statistically significant and the difference in patient types precludes direct comparison. However, these findings may suggest that if it were used for medium-sized gaps, the outcomes could be comparable.

Some studies reported complications [32,34,37], including the delayed union of the calcaneal bone block [32], infection [37], delayed healing of the incision, avulsed FHL transfer [14], heterotopic bone in the retrocalcaneal bursa, and fragmented calcaneal tuberosity with interosseus ossification proximal to the insertion [34].

Regardless of the technique used, the complication rate ranges from 0–21%, with most being minor complications [16]. The most frequent complications reported are related to the surgical wound [16,25]. In our series, we did not report any infections or wound problems. Even in the case where the allograft was used in the case of previous local infection (which was first debrided and treated with PMMA beads and antibiotics for 12 weeks), there were no new symptoms or wound issues and the allograft remained in good condition. There were no cases of re-rupture in any group. These results suggest that using allografts can be relatively safe when indicated.

Probably one of the most significant changes between the allograft and autologous techniques is the non-weight-bearing time. The allograft group has a greater time of non-weight bearing, until the correct union of the calcaneal bone block, usually between 8 and 12 weeks, compared to the autologous group, which could start weight bearing at the postoperative week 3. Table 4 summarizes the key distinctions between techniques.

Metabolic diseases, such as dyslipidemia, obesity, diabetes, and thyroid disorders, have been a focus in the treatment of Achilles tendon ruptures due to their association with poor postoperative outcomes. Oliva et al. [38] reported that 28% of the patients had a history of metabolic disease in acute Achilles tendon ruptures. In our sample of neglected chronic ruptures, 70.5% of the patients had at least one of these metabolic diseases.

The limitations of the present study include the retrospective design, the consequent loss of follow-up, and the small number of patients. The AOFAS_AH_ score was used to record the patient outcome, but AOFAS society recommended not to use this score anymore [39]. Additionally, the two groups are not directly comparable as they do not strictly treat the same tendon defect. A prospective well-powered study comparing similar patients and injury types would provide stronger evidence on this topic. However, the infrequent nature of chronic Achilles rupture presentation makes this difficult. This limitation might be addressed by conducting a multi-center study with a larger sample size.

Our study found comparable results in terms of function and scores in both groups, without the greater potential for complications in the allograft group. Therefore, we emphasize that reconstruction with an Achilles tendon allograft can be considered a viable treatment option, as it does not show a higher rate of complications. It could be a choice even for less extensive gaps, avoiding the soft tissue damage caused by a forced tendon advancement. However, further research is needed to fully understand the benefits and risks of each technique and to determine the best approach for treating chronic neglected Achilles tendon ruptures. The optimal approach may depend on the specific characteristics of the injury and the patient.

## 5. Conclusions

Chronic Achilles tendon ruptures can be successfully treated by careful selection of the reconstruction method according to the gap length and status of the remaining tissues. With an extensive defect, reconstruction with an Achilles tendon allograft should be considered a suitable treatment option as it does not present more complications than autologous techniques with comparable functional outcomes.

## Figures and Tables

**Figure 1 jcm-12-01135-f001:**
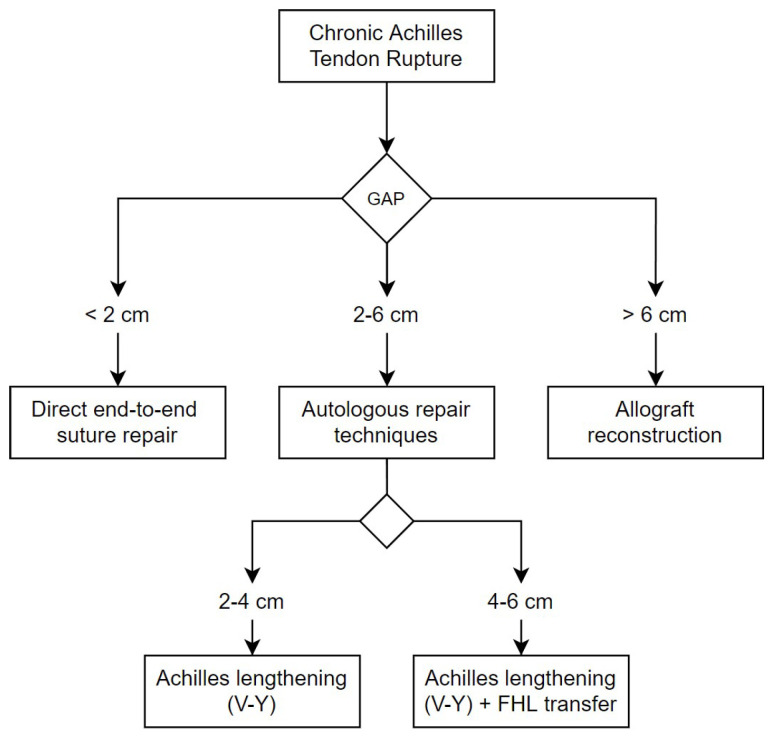
Our surgical algorithm for chronic Achilles tendon rupture, based on gap size. Lesser than 2 cm a direct end-to-end suture was performed. Between 2 cm and 6 cm, autologous techniques were used. If the gap was greater than 6 cm, reconstruction was done using an Achilles tendon allograft with calcaneal bone block.

**Figure 2 jcm-12-01135-f002:**
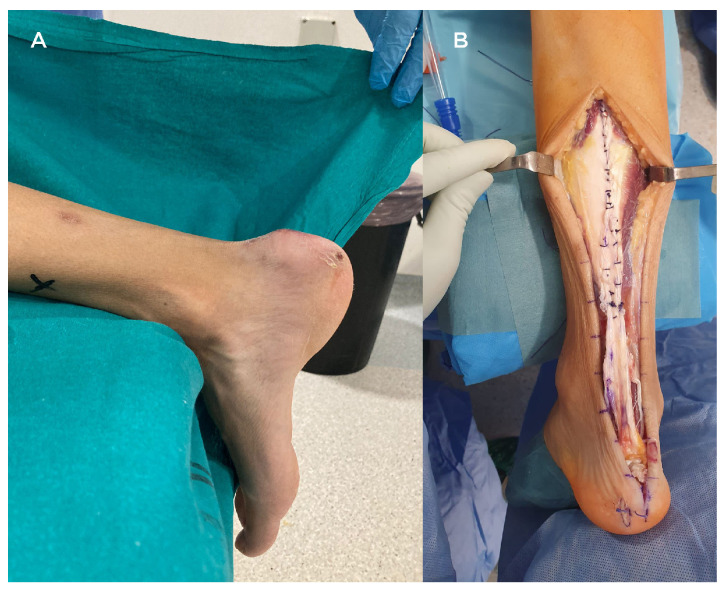
A case of a male patient with a less than 6 cm gap treated with V-Y tendon advancement and FHL transfer. (**A**) Preoperative foot exploration with visible sinking at the Achilles tendon. (**B**) An intraoperative picture with the technique performed: V-Y Achilles tendon advancement plus FHL transfer.

**Figure 3 jcm-12-01135-f003:**
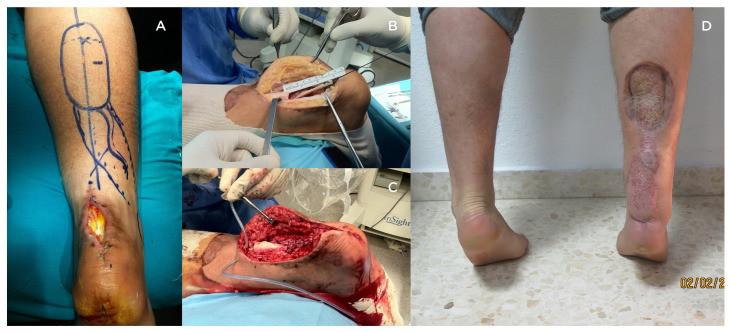
A case of absence of Achilles tendon secondary to an infection of an acute repaired rupture. The patient underwent initial surgery, which included deep debridement, gentamicin-PMMA bead implantation, and intravenous antibiotic therapy, to resolve the infection. Salvage reconstruction was then performed 12 weeks later. (**A**) A soft tissue defect is visible above the Achilles tendon, and a sural flap is planned for coverage. (**B**) Following resection of non-viable tissue, a 9 cm gap in the Achilles tendon was identified. A flexor hallucis longus (FHL) transfer was performed and sutured in place first. (**C**) Intraoperative photograph showing the sutured allograft filling the gap, in conjunction with the FHL transfer. (**D**) Long-term follow-up revealing functional heel-rise and the donor site of the sural flap covered with a skin graft.

**Table 1 jcm-12-01135-t001:** Subgroup data with intraoperative gap and type of surgery performed in each case.

Age	Gender	Gap (cm) ^1^	Group and Surgery
62	Male	3	Autologous techniques (V-Y advancement)
36	Male	3	Autologous techniques (V-Y advancement)
69	Male	3.5	Autologous techniques (V-Y advancement)
37	Female	3.5	Autologous techniques (V-Y advancement)
24	Male	4	Autologous techniques (V-Y advancement)
34	Male	4	Autologous techniques (V-Y advancement)
45	Male	5.5	Autologous techniques (V-Y + FHL transfer)
32	Male	6.5	Autologous techniques (V-Y + FHL transfer)
26	Male	6	Autologous techniques (V-Y + FHL transfer)
67	Female	7	Allograft reconstruction
32	Male	7	Allograft reconstruction
41	Male	7	Allograft reconstruction
56	Male	7	Allograft reconstruction
45	Male	8	Allograft reconstruction
47	Female	8	Allograft reconstruction
27	Male	9	Allograft reconstruction
48	Male	9	Allograft reconstruction (+FHL transfer)

^1^ Intraoperative measured gap.

**Table 2 jcm-12-01135-t002:** Demographic, patient characteristics, and follow-up data in global sample and subgroups.

	Autologous Techniques *n* = 9	Allograft Reconstruction *n* = 8	Global Sample *n* = 17	Significance
Sex (M/F)	8 M/1 F	6 M/2 F	14 M/3 F	0.00
Age (years)	40.55 ± 15.5 (24–69)	45.37 ± 12.68 (27–67)	42.82 ± 14.02 (24–69)	0.33
Metabolic disorders (%) *	6 (66.6%)	6 (75%)	12 (70.5%)	0.71
Gap (cms)	4.33 ± 1.32 (3–6.5)	7.75 ± 0.88 (7–9)	5.94 ± 2.07 (3–9)	0.00
Follow-up (months)	76.33 ± 36.33 (36–136)	88.75 ± 38.22 (26–150)	82.17 ± 36.60 (26–150)	0.42

* Obesity, hypercholesterolemia, diabetes mellitus, or thyroid disorders.

**Table 3 jcm-12-01135-t003:** Preoperative and postoperative ATRS and AOFAS_AH_ scores in the global and the different subgroups at the end of the follow-up, a mean of 82 ± 36.61 months (range 26–150). Results presented as median (interquartile range).

	Autologous Techniques *n* = 9	Allograft Reconstruction *n* = 8	Global Sample *n* = 17
ATRS pre	35 (33.75–38.5)	27 (19–45)	35 (24.5–40.5)
ATRS post	90 (85.5–95.25)	88 (79.5–94)	90 (79.75–95.25)
AOFAS_AH_ pre	55 (46–69)	50.5 (46.5–62)	55 (46–67)
AOFAS_AH_ post	96 (94–100)	95 (87–100)	96 (92.5–100)

**Table 4 jcm-12-01135-t004:** Summary of advantages and disadvantages of autologous vs. allograft techniques.

	Autologous Techniques	Allograft Reconstruction
Advantages	Suitable for small gaps Almost always available Shorter non-weight bearing time	Avoidance of donor site morbidity Smaller incision for large gaps Adequate tissue quantity Faster surgical procedure
Disadvantages	Greater incision and soft tissue stripping in large gaps Demanding technique in large gaps Increased surgical time Donor site morbidity	Infection concerns Potential immune response Increased cost Risk of no integration of bone block Longer non-weight-bearing time

## Data Availability

The data underlying this article are available from the corresponding author upon reasonable request.

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
