# Peer review of "Allograft and Autologous Reconstruction Techniques for Neglected Achilles Tendon Rupture: A Mid-Long-Term Follow-Up Analysis"

_jcm, 2023, doi:10.3390/jcm12031135_

Round 1

Reviewer 1 Report

This paper addressed an important and interesting problem- the efficiency of allograft and autologous reconstruction techniques for neglected Achilles tendon rupture. The authors compared allograft and autologous reconstruction techniques in 17 patients with Achilles tendon rupture regarding of the AOFAS score, ATRS score and complications. Overall, the article is well organized, and its presentation is good. However, some minor issues still need to be addressed:

(1) On page 1 line 12, please state the full name of abbreviations of AOFAS and ATRS.

(2) On page 1 line 14, this study advocated the reconstruction with Achilles tendon for the treatment. I suggest the cost, surgery time, hospitalization time and the length of incision be analyzed to further support the conclusion.

(3) On page 4 line 83, please state the statistical test for normality, and the power analysis for the minimum number of patients needed for each group was not performed.

(4) On page 4 line 81, this study reported the ability of performing a single-heel-rise was collected in the method part, but the consequence of this test was not presented in the section of result.

(5) AOFAS score was used for assessment of clinical outcome. However, the AOFAS society recommended not to use this score anymore (Pinsker & Daniels, FAI 2011, PMID 22097157).

(6) Inclusion criteria need to be more precise, such as age range, underlying disease conditions, etc. At the same time, I suggest that Table 1 reproduce the typesetting, which is convenient for the author to read.

(7) The authors are advised to add more research background on Chronic Achilles tendon ruptures, which helps to enhance the importance of highlighting the purpose of research. the authors might want to consider including some recent work. (e.g., doi:10.1111/os.12429, doi: 10.1007/s00167-019-05723-9; 10.1186/s12891-016-0967-1). Moreover, authors are advised to include relevant biological material content in the Discussion. (e.g., doi: 10.1038/s41551-021-00810-0; 10.1016/j.bioactmat.2022.01.033; 10.1016/j.actbio.2022.03.033)

I would be very glad to re-review the paper in greater depth once it has been edited because the subject is interesting.

Author Response

The manuscript has been improved for the evaluated points, and the text has been revised for improved readability with a full grammar revision.

This paper addressed an important and interesting problem- the efficiency of allograft and autologous reconstruction techniques for neglected Achilles tendon rupture. The authors compared allograft and autologous reconstruction techniques in 17 patients with Achilles tendon rupture regarding of the AOFAS score, ATRS score and complications. Overall, the article is well organized, and its presentation is good. However, some minor issues still need to be addressed:

(1) On page 1 line 12, please state the full name of abbreviations of AOFAS and ATRS. 

These abbreviations have been added

(2) On page 1 line 14, this study advocated the reconstruction with Achilles tendon for the treatment. I suggest the cost, surgery time, hospitalization time and the length of incision be analyzed to further support the conclusion.

We have corrected the text to include more details on these points and to clarify that the reconstruction with Achilles tendon is not presented as the best option, but as a possible solution when gaps are very large. We agree that the cost, surgery time, hospitalization time and the length of incision should be analyzed to further support our conclusion, and these comments were added in the discussion section

(3) On page 4 line 83, **please state the statistical test for normality, and the power analysis for the minimum number of patients needed for each group was not performed.

We have detailed the statistical test for normality, which was performed using the Shapiro-Wilk test. All continuous data were found to be statistically normal. We treated scoring systems as nonparametric due to their nature and the small sample size. We also corrected the methods and results exposition to be presented more accurately.

Understanding that this is not a clinical trial, and was performed in a retrospective manner, we did not perform a priori power analysis or sample size calculation. With regards to the post-hoc power analysis, it is acknowledged in the literature that it is not a recommended practice, especially in retrospective studies for being analytically misleading. (10.1097/SLA.0000000000002908, 10.1136/gpsych-2019-100069)

(4) On page 4 line 81, this study reported the ability of performing a single-heel-rise was collected in the method part, but the consequence of this test was not presented in the section of result**.

The ability of performing a single-heel-rise was included in the results part.

(5) AOFAS score was used for assessment of clinical outcome. However, the AOFAS society recommended not to use this score anymore (Pinsker & Daniels, FAI 2011, PMID 22097157).

We have included this reference to the AOFAS society's recommendation in the manuscript as acknowledgement of this. We understand that the AOFAS society no longer recommends this score for clinical outcomes assessment. However, as it was the score initially recorded and the data is based on this, we feel it is fair to include it in our analysis. Despite this, we will make sure to consider this recommendation in future studies.

(6) Inclusion criteria need to be more precise, such as age range, underlying disease conditions, etc. 

We have updated the age range in the inclusion criteria. Initially, we did not establish many exclusion criteria as neglected Achilles tendon ruptures are not a common injury. We have made the necessary adjustments to provide a more precise description of the patients included in the study. Also, metabolic disease condition data was added.

At the same time, I suggest that Table 1 reproduce the typesetting, which is convenient for the author to read.

We have made further corrections to the tables to be better aligned with the desired format. We did an additional effort to ensure that the manuscript is typeset in the scientific preferred LaTeX format, which also facilitates post-processing in preparation for publication. We have meticulously followed the MDPI guidelines for layout using its template. Some minor issues are impossible to be modified in the LaTeX template, but we trust that the MDPI's can address them as they states in their styling guide, "we pride ourselves on providing a comprehensive production service prior to publication."

(7) The authors are advised to add more research background on Chronic Achilles tendon ruptures, which helps to enhance the importance of highlighting the purpose of research. the authors might want to consider including some recent work. (e.g., doi:10.1111/os.12429, doi: 10.1007/s00167-019-05723-9; 10.1186/s12891-016-0967-1). 
Moreover, authors are advised to include relevant biological material content in the Discussion. (e.g., doi: 10.1038/s41551-021-00810-0; 10.1016/j.bioactmat.2022.01.033; 10.1016/j.actbio.2022.03.033)

We have updated the references and included some recent articles in the discussion section that were not added in the first manuscript.

However, after reading thoroughly the suggested references regarding the biological content, we do not see a clear connection to the primary topic of our study, in terms of the comparison of the techniques for the treatment of chronic neglected Achilles tendon ruptures. We appreciate the importance of including relevant research in our discussion but humbly believe that these references do not add value to our specific study and should not be included in order to maintain the focus of our research.

I would be very glad to re-review the paper in greater depth once it has been edited because the subject is interesting.

We would like to express our sincerest gratitude for taking the time to review our manuscript and providing valuable feedback and suggestions. Your insights have been extremely helpful and we have taken them into consideration in revising our work. We understand the importance of thorough review in the publishing process and we are grateful for your expertise. We hope that the revisions made to the manuscript meet your expectations and we look forward to your continued guidance.

Reviewer 2 Report

Many thanks to the authors for having presented a so interesting paper Allograft and autologous reconstruction techniques for neglected Achilles tendon rupture: a mid-long-term follow-up analysis.

Please before resubmitting the revision version of the article, read the editorial rules carefully, and check for other editorial aspects, such as: text alignment, text justification at the head, etc.

The language is good that the manuscript does not need to be corrected by a person of English mother tongue, but it only needs a fast grammar and syntax revision of some sentences.

Abstract

The abstract is well structured, and it contains the main information of the study.

Key words

Please provide them in alphabetic order.

Background

 The introduction is quite well structured, containing the main aims of the study.

Line 20: "It is more common in the middle-aged population that participate occasionally in sports activities [2,3]". Please add other risk factors of achilles tendon ruptures.

Line 25: Many surgical procedures have been proposed and claimed to have good clinical outcomes.

Please quote the most popular percutaneous procedure, adding the following reference:

·      The repair of the Achilles tendon rupture: comparison of two percutaneous techniques

Strategies Trauma Limb Reconstr. 2011 Nov;6(3):147-54.  doi: 10.1007/s11751-011-0124-1. Epub 2011 Nov 8. PMID: 22065368  PMCID: PMC3225567  DOI: 10.1007/s11751-011-0124-1

Methods

This section contains enough information to understand and possibly repeat the study. 

Results

The results presented are complete; but please introduce some tables to graphically represent the results of the study

Statistical analysis

The statistical analysis is accurate and detailed.

Discussion

The discussion is detailed and explanatory, but I find it appropriate to summarize a few paragraphs and introduce summary tables regarding the advantages and disadvantages of the techniques described. Further, discuss also metabolic aspect with your results, quoting.

·      Achilles Tendon Rupture and Dysmetabolic Diseases: A Multicentric, Epidemiologic Study. J Clin Med. 2022 Jun 27;11(13):3698. doi: 10.3390/jcm11133698.

Conclusions

The conclusions reflect and refer to the results of the study, despite the low sample size.

References

The references are not up to date. However, please delete those before 2010 (1,2,4,5,11,16,17,19,22,29,33) if not strictly essential, eventually replacing them with newer ones and integrate them with those suggested previously. 

Tables and Figures

The number and quality of tables are appropriate to transmit the main information of the paper.

Author Response

Thank you for your time and suggestions on our manuscript. The manuscript has been improved for the evaluated points, and the text has been full revised for improved readability.

Many thanks to the authors for having presented a so interesting paper Allograft and autologous reconstruction techniques for neglected Achilles tendon rupture: a mid-long-term follow-up analysis.

Please before resubmitting the revision version of the article, read the editorial rules carefully, and check for other editorial aspects, such as: text alignment, text justification at the head, etc.

We have made further corrections to the article to be better aligned with the desired format. We did an additional effort to ensure that the manuscript is typeset in the scientific preferred LaTeX format, which also facilitates post-processing in preparation for publication. We have meticulously followed the MDPI guidelines for layout using its template, but some minor formatting issues may arise when working with LaTeX. In this regard, as MDPI states in their styling guide, "we pride ourselves on providing a comprehensive production service prior to publication.", we trust that the MDPI's can address these minor issues during the post-processing phase as some are impossible to be modified in the LaTeX template.

The language is good that the manuscript does not need to be corrected by a person of English mother tongue, but it only needs a fast grammar and syntax revision of some sentences.

The manuscript has been revised for improved readability with a full grammar revision.

Abstract
The abstract is well structured, and it contains the main information of the study. 

Key words
Please provide them in alphabetic order.

Keywords were changed to appear in alphabetic order

Background
The introduction is quite well structured, containing the main aims of the study.

Line 20: "It is more common in the middle-aged population that participate occasionally in sports activities [2,3]". Please add other risk factors of achilles tendon ruptures.

We have added additional information on risk factors for achilles tendon ruptures. We have also included an updated reference to support this information.

Line 25: Many surgical procedures have been proposed and claimed to have good clinical outcomes.
Please quote the most popular percutaneous procedure, adding the following reference:
·      The repair of the Achilles tendon rupture: comparison of two percutaneous techniques Strategies Trauma Limb Reconstr. 2011 Nov;6(3):147-54.  doi: 10.1007/s11751-011-0124-1. Epub 2011 Nov 8. PMID: 22065368  PMCID: PMC3225567  DOI: 10.1007/s11751-011-0124-1

We understand the importance of including information on various surgical procedures for Achilles tendon ruptures in our article. However, our main focus is on comparing surgical techniques for major tendon defects in chronic ruptures, where these percutaneous techniques are not used. The suggested article doesn't mention chronic ruptures. Therefore, we humbly believe that this reference is not in consistency with our research objectives.

Methods
This section contains enough information to understand and possibly repeat the study. 

Results
The results presented are complete; but please introduce some tables to graphically represent the results of the study.

Tables have been updated and corrected in format. A table has been added with a summary of the results for improved exposition of the data as suggested.

Statistical analysis
The statistical analysis is accurate and detailed.

Discussion
The discussion is detailed and explanatory, but I find it appropriate to summarize a few paragraphs and introduce summary tables regarding the advantages and disadvantages of the techniques described. Further, discuss also metabolic aspect with your results, quoting.

·      Achilles Tendon Rupture and Dysmetabolic Diseases: A Multicentric, Epidemiologic Study. J Clin Med. 2022 Jun 27;11(13):3698. doi: 10.3390/jcm11133698.

A summary table has been included that highlights the advantages and disadvantages of the techniques described in the study. Additionally, a paragraph addressing the metabolic aspect of Achilles tendon ruptures has been added in the discussion section. The suggested citation is added. We also reviewed the presence of these conditions in our sample.

Conclusions

The conclusions reflect and refer to the results of the study, despite the low sample size.

References

The references are not up to date. However, please delete those before 2010 (1,2,4,5,11,16,17,19,22,29,33) if not strictly essential, eventually replacing them with newer ones and integrate them with those suggested previously. 

The non essential and old references have been removed, the list is now updated.

Tables and Figures

The number and quality of tables are appropriate to transmit the main information of the paper.

We would like to express our sincerest gratitude for taking the time to review our manuscript and providing valuable feedback and suggestions. Your insights have been extremely helpful and we have taken them into consideration in revising our work. We understand the importance of thorough review in the publishing process and we are grateful for your expertise. We hope that the revisions made to the manuscript meet your expectations and we look forward to your continued guidance.

Round 2

Reviewer 2 Report

In the introduction section, a few lines about general surgical repair of Achilles tendon ruptures should be added, in particular about percutaneous techniques, quoting:

·         The repair of the Achilles tendon rupture: comparison of two percutaneous techniques.  Strategies Trauma Limb Reconstr.  2011 Nov;6(3):147-54. doi: 10.1007/s11751-011-0124-1.

Author Response

In the introduction section, a few lines about general surgical repair of Achilles tendon ruptures should be added, in particular about percutaneous techniques, quoting:

  • The repair of the Achilles tendon rupture: comparison of two percutaneous techniques.  Strategies Trauma Limb Reconstr. 2011 Nov;6(3):147-54. doi: 10.1007/s11751-011-0124-1.

Thank you for your feedback and suggestions on our manuscript. Few lines about general surgical repair of Achilles tendon ruptures has been added as well as the suggested reference.